# Can the Spontaneous Electroencephalography Theta/Beta Power Ratio and Alpha Oscillation Measure Individuals’ Attentional Control?

**DOI:** 10.3390/bs14030227

**Published:** 2024-03-12

**Authors:** Hua Wei, Lele Chen, Lijun Zhao

**Affiliations:** 1Department of Psychology, Suzhou University of Science and Technology, Suzhou 215000, China; weihuaweihua@mail.usts.edu.cn; 2Department of Applied Psychology, School of Education Science, Nantong University, Nantong 226000, China; 3Department of Psychology, School of Education Science, Liaocheng University, Liaocheng 252000, China

**Keywords:** spontaneous EEG, theta/beta power ratio, alpha oscillation, attentional control, flanker

## Abstract

Past studies have shown that spontaneous electroencephalography indicators—namely, the theta/beta power ratio and alpha oscillation—may measure individuals’ attentional control processes. However, there is lack of research distinguishing these differences. This study investigated whether the theta/beta power ratio and alpha oscillation were separately related to the objective and subjective criteria of attentional control in eyes-open and eyes-closed conditions. The results showed two main findings: (1) In the eyes-open condition, the theta/beta power ratio at the Fz and Pz electrode sites were significantly negatively correlated with the attentional control scale score; the alpha power at the Pz electrode site was significantly negatively correlated with flanker RT interference effect; (2) In the eyes-closed condition, the alpha power at the Cz and Pz electrode sites were significantly positively correlated with flanker P3d. In summary, this study showed that the eyes-open spontaneous theta/beta power ratio may reflect individuals’ beliefs in their attentional control ability, and the alpha oscillation may be related to individuals’ attentional control ability.

## 1. Introduction

Attentional control is the cognitive ability to regulate attention by selectively focusing on relevant information while ignoring distractions; this ability is essential to achieve specific tasks and goals. It is a key component in central executive functions and provides the foundation for basic cognitive processes, such as attention and memory [1,2]. Deficits in attentional control have been identified as a central feature of several mental and psychological disorders that affect cognitive function [3,4,5,6,7]. Therefore, effective measure of attentional control plays a crucial role in advancing research in related fields.

Traditional attentional control measures primarily rely on a specific cognitive task and scale [8,9,10]. Because attentional control cannot be directly measured, common inhibitory control tasks that were used to measure individuals’ objective attentional control ability include the flanker, Stroop, and Go/Nogo tasks [8]. Derryberry and Reed [10] developed the attentional control scale (ACS), a self-report scale that can effectively measure individuals’ subjective attentional control beliefs. Moreover, in recent years, a growing body of research has pointed to the feasibility of spontaneous electroencephalography (EEG) to measure individuals’ attentional control. It has been suggested that, without relying on specific measures, spontaneous EEG can provide a possible reflection of individuals’ attentional control processes [11].

The most commonly used indicator is the theta/beta power ratio in spontaneous EEG, which may serve as a good biomarker for individuals’ attentional control beliefs: a higher theta/beta power ratio indicates a weaker belief in one’s own attentional control [11,12,13,14,15]. In addition, the alpha oscillation in spontaneous EEG may be related to attentional control ability. Previous studies have shown that an increase in alpha power may reflect enhanced inhibitory control over task-irrelevant regions of the brain, while a decrease in alpha power may reflect an increase in attention resources engaged in the current task [16,17]. However, existing studies have produced conflicting results regarding the effectiveness of these two indicators for measuring individuals’ attentional control, and there is lack of research distinguishing the two indicators [14,18,19].

Putman, Van Peer, et al. [11] conducted a study of 28 healthy participants and found that the individuals’ theta/beta power ratios were inversely correlated with their ACS scores (*r* = −0.395, *p* = 0.037), providing preliminary evidence for a possible association between the theta/beta power ratio and individuals’ attentional control beliefs. Putman, Verkuil, et al. [12] replicated this finding in a study of 77 healthy participants (*r* = –0.33, *p* = 0.004). Angelidis, Der Does, et al. [13] collected spontaneous EEG data and ACS scores from 41 participants over a one-week period and found that the theta/beta power ratio’s test–retest reliability was 0.93, and the ratio significantly predicted ACS scores one week later (*r* = −0.44, *p* = 0.014). In summary, these studies support the possibility that the theta/beta power ratio may serve as a marker of individuals’ attentional control beliefs, as a higher theta/beta power ratio indicates a weaker belief in one’s attentional control ability.

However, some studies have failed to support this notion. Angelidis, Hagenaars, et al. [14] did not find a significant correlation between the theta/beta power ratio and individual ACS scores in a study of 65 participants. In addition, Morillas-Romero, Tortella-Feliu, et al. [18] did not find a significant correlation between this ratio and individuals’ ACS scores in a study of 107 participants and instead found that the theta/beta power ratio was not significantly correlated with flanker task performance. Similarly, Zhang, Li, et al. [20] used the stop-signal task to measure individuals’ attentional control and found no significant correlation between task performance and the theta/beta power ratio. These above results do not support the hypothesis that the theta/beta power ratio can serve as a marker of individuals’ attentional control beliefs, but do show that this indicator is not related to attentional control ability.

While some studies have demonstrated a correlation between spontaneous EEG alpha oscillations and attention activity, there is a lack of clear evidence supporting a correlation between spontaneous EEG alpha activity and attentional control ability [21,22,23]. The popular idling hypothesis is that alpha oscillation is an inverse marker of arousal; that is, alpha power decreases in the eyes-open state compared to that in the eyes-closed state [17]. According to this hypothesis, a decrease in alpha power reflects an individual’s increased attention to the current task. Meanwhile, in recent years, the opposite view has been suggested—namely, that alpha oscillations may be related to inhibition and that an increase in alpha band power in the parietal lobe may reflect an increase in the amount of attention resources being engaged to enhance inhibitory control over task-irrelevant regions in the brain [16,23,24]. However, evidence supporting the latter, more recent hypothesis is not based on studies on spontaneous alpha EEG.

Therefore, the present study investigated whether two spontaneous EEG indicators (i.e., theta/beta power ratio and alpha oscillation) can serve as effective measures of individuals’ attentional control. To address this question, we needed to consider the following three aspects. First, we considered the difference between spontaneous EEG in eyes-open and eyes-closed conditions. Most spontaneous EEG power spectrum analyses in previous related studies did not distinguish between these two conditions. Whether the eyes are open significantly influences the magnitude of alpha oscillation. For example, eyes being open is frequently linked to heightened sensory input and increased attention, whereas closed eyes are commonly associated with a comparatively more relaxed state. Individuals in the eyes-open condition always have smaller alpha power than in eyes-closed condition [17]. And the reproducibility of spontaneous EEG in eyes-closed conditions is slightly higher than that in eyes-open conditions [25]. Thus, we conducted a power spectrum analysis while distinguishing between the two conditions. Second, we needed to focus on the effectiveness of attentional control measures. Previous research commonly used behavioral tasks or ACS to assess attentional control; some studies suggested that attentional control behavioral tasks measure attentional control ability, while ACS only measures beliefs about individuals’ attentional control ability [26,27]. In particular, when individuals perform a behavioral task, event-related potentials (ERPs) may provide more intuitive measures of attentional control ability. In the present study, the flanker task was used to measure individuals’ attentional control ability. The flanker-related ERP components of N2 and P3 are considered to be associated with attentional control ability. A larger N2 amplitude, especially N2d amplitude (incongruent N2 amplitude minus congruent N2 amplitude), may reflect an increasing response conflict while performing the flanker task [28,29,30]. However, a smaller P3 amplitude, especially P3d amplitude (incongruent P3 amplitude minus congruent P3 amplitude), may reflect that fewer available attentional resources can be used to perform the flanker task [31]. A larger P3 amplitude, especially P3d amplitude, reflects that enhanced attentional resources were used to complete the flanker task [28,32,33,34]. Hence, this study incorporated measures of attentional control through both a flanker task and questionnaire items as reference standards for calibration.

In summary, the present study investigated whether spontaneous EEG indicators (theta/beta power ratio and alpha oscillation) can measure attentional control (belief or ability). To solve this problem effectively, we consider the following two aspects: (1) two spontaneous conditions (eyes-open and eye-closed), and (2) objective and subjective criteria of attentional control (flanker task and ACS). We hypothesized that the theta/beta power ratio may be associated with beliefs about attentional control, while alpha oscillations may be linked to actual attentional control abilities.

## 2. Materials and Methods

### 2.1. Participants

In previous research [11], individuals’ theta/beta power ratios were inversely correlated with their ACS scores (*r* = −0.395). We set α = 0.05, power = 0.8, and set the effect size at 0.395; G*Power recommended a minimum total sample size of 45 for correlation analysis. Fifty-eight participants (age range: 18–23 years, 46 women) recruited from Liaocheng University were invited to participate in the study. Before the experiment, all participants provided written informed consent and were informed of their right to withdraw from the study at any time. The Ethics Committee of the Suzhou University of Science and Technology approved the experimental procedures, and the study was conducted in accordance with the approved procedures.

### 2.2. Experimental Materials

#### 2.2.1. Subjective Attentional Control Measure: ACS

The ACS is a 20-item self-report measure that was developed by Derryberry and Reed, which is used to assess attentional control; in particular, it measures individuals’ beliefs in their ability to control their own attention [10,27]. The Chinese version of ACS was translated and revised by Huizi Zhang, building upon the English version [35]. This version exhibits robust reliability and validity (Cronbach’s alpha: 0.825, n = 420) [35]. The present study used the Chinese version of ACS.

#### 2.2.2. Objective Attentional Control Measure: Flanker Task

The flanker task is a behavioral measure used to assess individuals’ attentional control ability. It comprises both congruent and incongruent conditions. A larger difference in response time (RT) between the congruent and incongruent conditions is generally believed to indicate a larger conflict. Similar to previous studies [36,37], as shown in Figure 1, for each trial, a fixation cross (“+”) was displayed in the middle of the computer screen (for 300 ms), marking the beginning of the experimental trial; then, after waiting with a blank screen (for 1200–1600 ms, randomized), five arrows appeared with a 200 ms presentation time as the flanker stimulus. The participants were positioned approximately 70 cm from a 21-inch computer screen. The arrows displayed on the screen subtended a visual angle of 1.3° vertically and 1.3° horizontally, with a distance of 0.25° between them. The inter-trial interval, randomized for 2000–2400 ms, separated each trial. The stimulus appeared in white against a black screen.

### 2.3. Experimental Procedure

The experiment consisted of three steps. First, participants were required to use a smartphone to complete the questionnaire (in approximately 15 min). Second, after preparation for the EEG experiments (for approximately 45 min), each participant received an 8 min spontaneous EEG recording, during which they were instructed to (1) fixate on a cross (“+”) displayed at the center of a computer screen and (2) alternate between opening and closing their eyes every minute. Third, participants performed the flanker task while EEG data were recorded (for about 20 min). The present arrow version of the flanker task consisted of one practice block (10 congruent and 10 incongruent trials) and two formal blocks (60 congruent and 60 incongruent trials in each block). There were two different experimental conditions (50% chance for each condition): (1) in the congruent condition, the target and distractor arrows had the same orientation, and (2) in the incongruent condition, the target and distractor arrows had opposite orientations. The participants were instructed to identify the direction of the target arrow with both speed and precision: they had to press the ‘‘F’’ button if it pointed left or the ‘‘J’’ button if it pointed right on the computer keyboard. After completing the experiment, the participants were remunerated appropriately.

### 2.4. EEG Data

#### 2.4.1. EEG Data Collection

EEG recordings were obtained using the ANT Neuro system: 64 scalp electrodes were placed according to the international 10–20 system, with a pass-band of 0.01–100 Hz and a sampling rate of 1000 Hz. A 24-bit resolution was used for data acquisition. Each participant was seated comfortably in a separate room, with AFz serving as the ground lead and CPz as the reference during the EEG data recording. Impedances were checked before recording and maintained below 10 kΩ.

#### 2.4.2. EEG Data Analysis

The EEG data were processed using the open-source toolbox EEGLAB (version: 14_1_2_b), which runs within the MATLAB environment (version: 2018b). To prepare the data for analysis, a 30 Hz low-pass filter and a 0.5 Hz high-pass filter were applied to the continuous EEG data. And EEG data was also re-referenced to the average mastoids for further analysis.

Spontaneous EEG data were segmented into 1000 ms epochs. Any data that did not meet the standard quality control measures were modified or removed a priori. First, any data from trials with large drifts were manually removed. Second, data from trials contaminated by eyeblinks were modified using an independent components analysis algorithm (infomax). The spontaneous EEG data with the eyes-open and eyes-closed conditions were divided into two different datasets. To examine the frequency distribution of the EEG signals, we applied a fast Fourier transform (Welch algorithm) with no phase shift and a frequency resolution of 0.9766 Hz. This transformed the EEG signals into the frequency domain, allowing us to obtain the EEG spectral power in the range of 1–30 Hz. We calculated the absolute power (μV^2^) for the theta (4–8 Hz), alpha (8–13 Hz), and beta (13–30 Hz) frequency bands. Given that neural activity in different brain regions may undertake diverse roles in the attentional control process, and based on prior research [15,38], we collected the absolute power obtained from the frontal (Fz), central (Cz), and parietal (Pz) electrodes for further statistical analysis. And, to calculate the theta/beta power ratio, we divided the theta power by the beta power at each electrode site. To facilitate additional statistical analysis and ensure data normalization, we applied square root transformations to the absolute alpha power and the theta/beta power ratio.

To extract the ERP data for flanker-related EEG signals, EEG epochs were extracted using a window from −200 to 1000 ms, which was time-locked to the stimulus onset. Then, to remove any potential baseline effects, the EEG data were baseline-corrected using the pre-stimulus interval. As with the analysis of the spontaneous EEG data, standard quality control measures were applied to the data. In addition, the data from incorrect trials were removed. Any data that did not meet these criteria were either modified or removed before further analysis. We followed prior research [31,36] and performed a visual inspection of ERP waveforms and topographical maps to extract flanker-related N2 and P3 amplitudes. In the present study, a visual inspection of topographical maps revealed pronounced increases in N2 wave at electrode Fz and P3 wave at CPz. Accordingly, for further statistical analysis, we quantified the N2 amplitude as the negative mean amplitude at Fz between 230 and 380 ms after stimulus onset and the P3 amplitude as the positive mean amplitude at CPz between 400 and 600 ms after stimulus onset.

### 2.5. Statistical Analysis

We conducted statistical analysis using JASP software (version: 0.18.1.0), which is available at https://jasp-stats.org (accessed on 31 October 2023). The RT interference effect, N2d (N2 amplitude interference effect), and P3d (P3 amplitude interference effect) were calculated by subtracting the RTs, N2 amplitude, and P3 amplitude in the incongruent condition from those in the congruent condition, respectively. We used two-tailed Pearson’s correlation analyses to examine the correlations between the theta/beta power ratio or alpha power and the subjective measures (ACS) or objective attentional control measures (RT interference effect, N2d, and P3d). All data are reported as mean ± standard error (M ± SE).

## 3. Results

### 3.1. Preliminary Analyses

The results were as follows. (1) The incongruent trials had slower RTs (528.74 ± 10.59 ms) than the congruent trials (466.29 ± 10.48 ms; t(57) = 20.77, *p* < 0.001, Cohen’s d = 2.73). (2) The incongruent trials had lower accuracy (0.9253 ± 0.0151) than the congruent trials (0.9761 ± 0.0070; t(57) = 4.41, *p* < 0.001, Cohen’s d = 0.58. (3) The incongruent trials had larger N2 amplitude (−2.59 ± 0.63 µV) than the congruent trials (−1.54 ± 0.63 µV; t(57) = 4.49, *p* < 0.001, Cohen’s d = 0.59), as shown in Figure 2. (4) The incongruent trials had a larger P3 amplitude (7.30 ± 0.57 µV) than the congruent trials (6.23 ± 0.53 µV; t(57) = 3.12, *p* = 0.003, Cohen’s d = 0.41), as shown in Figure 2.

### 3.2. Correlation between the Theta/Beta Power Ratio and Subjective/Objective Attentional Control Measures

As shown in Table 1, paired-sample *t*-tests indicate that participants in the eyes-open condition had smaller theta/beta power ratio than in the eyes-closed condition at the Fz electrode site (t(57) = 2.68, *p* = 0.01, Cohen’s d = 0.35) and Pz electrode site (t(57) = 2.20, *p* = 0.03, Cohen’s d = 0.29), but not at Cz electrode site (t(57) = 1.56, *p* = 0.13, Cohen’s d = 0.21).

#### 3.2.1. Eyes-Open Condition

As shown in Table 2, the results indicate that the theta/beta power ratio was significantly negatively correlated with the ACS score at Fz electrode (*p* = 0.046) and Pz electrode sites (*p* = 0.046). No other significant correlations were found.

#### 3.2.2. Eyes-Closed Condition

The results revealed no significant correlations; see Table 2.

### 3.3. Correlation between Alpha Power and Subjective/Objective Attentional Control Measures

As shown in Table 1, paired-sample t-tests indicate that participants in the eyes-open condition had smaller alpha power than in eyes-closed condition at the Fz electrode site (t(57) = 10.19, *p* < 0.001, Cohen’s d = 1.34), Cz electrode site (t(57) = 10.63, *p* < 0.001, Cohen’s d = 1.40), and Pz electrode site (t(57) = 9.64, *p* < 0.001, Cohen’s d = 1.27).

#### 3.3.1. Eyes-Open Condition

As shown in Table 3, the results revealed that alpha power was significantly negatively correlated with RT interference effect at the Pz electrode site (*p* = 0.044).

#### 3.3.2. Eyes-Closed Condition

As shown in Table 3, the results were as follows. The absolute alpha power was significantly positively correlated with P3d at the Cz electrode (*p* = 0.026) and Pz electrode sites (*p* = 0.03).

### 3.4. Correlation between Attentional Control Measures

As shown in Table 4, the results indicated that ACS scores were significantly positively correlated with the RT interference effect (*p* = 0.02); the RT interference effect was significantly negatively correlated with the P3d (*p* = 0.047); and the N2d was significantly positively correlated with the P3d (*p* = 0.001). No other significant correlations were found.

## 4. Discussion

To the best of our knowledge, this study represents the first direct investigation into the relationship between two spontaneous EEG indicators (theta/beta power ratio and alpha oscillation) and individuals’ attentional control in eyes-open and eyes-closed conditions separately. The findings showed that (1) in the eyes-open condition, the theta/beta power ratio was negatively correlated with the ACS score at Fz electrode and Pz electrode sites; the alpha power was negatively correlated with RT interference effect at the Pz electrode site; (2) in the eyes-closed condition, the alpha power was positively correlated with the P3d at the Cz and Pz electrode sites. Our study validates an association between the theta/beta power ratio and attentional control beliefs, as well as a correlation between alpha oscillations and attentional control ability.

Individuals’ subjective beliefs about attentional control ability and objective attentional control ability are related but distinct constructs [26,27]. Our research also supports this notion, as the results showed that the ACS was moderately correlated with the RT interference effect but not with N2d or P3d. Thus, the present study may help explain why previous studies failed to find a correlation between the theta/beta power ratio and objective measures of attentional control ability despite observing associations between the theta/beta power ratio and ACS scores, response inhibition, and attentional bias for emotional stimuli. Some prior studies confirmed that the theta/beta power ratio was corelated negatively with ACS scores [11,13]. For example, Putman, Verkuil, et al. [12] found that the theta/beta power ratio accounted for 28% of the variability in the decline in self-reported attentional control under stress. Additionally, Putman, Van Peer, et al. [11] found that a higher theta/beta power ratio was associated with poorer inhibitory control over fearful facial stimuli, which provided evidence that the theta/beta power ratio modulates individuals’ attentional processing of emotional information. Further investigation revealed that individuals with a high theta/beta power ratio showed attentional facilitation toward moderately threatening stimuli and attentional avoidance toward highly threatening stimuli, compared with individuals with a low theta/beta power ratio [14]. Only individuals with a low theta/beta power ratio could allocate attentional control resources effectively, reducing attentional resource allocation to moderately threatening stimuli and coping with highly threatening stimuli [14]. Thus, the above studies suggest that the theta/beta power ratio may serve as a marker of attentional control belief, and a low theta/beta power ratio indicates high attentional control belief.

However, it is insufficient to conclude that the theta/beta power ratio is an effective marker of individuals’ attentional control ability based solely on the aforementioned research results. Although past studies showed that the theta/beta power ratio is related to inhibition and attentional bias toward emotional stimuli [11,14], currently, there is no direct evidence supporting a relationship between the theta/beta power ratio and individual performance in attentional control-related tasks [18,20]. Our research supports the association between the theta/beta power ratio and individuals’ beliefs regarding attentional control; however, it demonstrates no significant association between the ratio and attentional control ability. The present study can help explain why previous studies failed to find a correlation between the theta/beta power ratio and measures of attentional control ability.

In addition, although some studies found a negative correlation between the theta/beta power ratio and ACS scores [11,13], others have not supported this conclusion [14,18]. Our study contributes to understanding of the inconsistent outcomes, as it demonstrated that the theta/beta power ratio specifically relates to individuals’ beliefs regarding attentional control only during the eyes-open condition. The results showed that participants in the eyes-open condition had smaller theta/beta power ratio than in eyes-closed condition. However, in these previous studies, the theta/beta power ratio was always calculated by combining open- and closed-eye data [11,12,13,18]. The inconsistent and unexplained results in previous research may be largely attributed to the failure to differentiate between eyes-open and eyes-closed conditions. That is, during the data analysis process, failure to distinguish between the effects of eyes open and those of eyes closed on spontaneous EEG activity may lead to erroneous conclusions. Of course, this explanation is only a possibility because the correlation between the theta/beta power ratio and ACS scores was not strong in this study. The present study offers limited insights into the mechanism of the theta/beta power ratio. For instance, if theta/beta power ratio does not accurately reflect attentional control but indicates attention orientation, a reassessment of previous findings is needed. In Morilla-Romero et al.’s study [18], the theta/beta ratio showed no significant correlation with ACS scores or executive control ability in the Attentional Network Test. Surprisingly, a significant negative correlation was found with individual attention orientation (*r* = −0.217, *p* = 0.03), prompting a reconsideration of the theta/beta ratio’s role in attentional orientation abilities. Thus, further confirmation of this point will be essential.

Interpretation of the P3 amplitude in the flanker task has been a subject of contentious debate in the academic literature [28,32,33,34,39]. In the flanker task, a larger RT interference effect suggests constrained attentional resources, which may impede the completion of the current task and lead to a heightened level of conflict [36,40]. Our findings suggest that individuals exhibit a greater P3 amplitude when faced with incongruent stimuli compared with congruent stimuli. And the reduction in the RT interference effect is accompanied by a decrease in P3d. These findings support the notion that the P3 amplitude serves as a measure of individual attentional resources utilized in the execution of the flanker task, whereby increased task difficulty demands greater attention resources, leading to a larger P3 amplitude. Additionally, our study identified a significant positive correlation between the spontaneous alpha power in the eyes-closed condition and P3d amplitude, providing evidence for the inhibition-timing hypothesis [16]. Specifically, an increase in the alpha band power, especially within the central and parietal lobe, may indicate an increase in attention resources employed during the task at hand. In essence, our findings illustrated that an increase in spontaneous alpha power signified an enhanced ability for attentional control in individuals. Consequently, they exhibited an increased involvement of attentional resources during the completion of the flanker task, as evidenced by the heightened P3d. Additionally, a reduction in conflict was observed, as indicated by the decreased RT interference effect. However, as no correlation between alpha and N2d amplitude was identified in the flanker task. Further research is warranted to ascertain the feasibility of employing alpha as a marker of individuals’ attentional control ability.

Nevertheless, it is crucial to acknowledge that the limitations in this study may impact the reliability and validity of the conclusions. In previous research [11], G*Power recommended a minimum total sample size of 45 for correlation analysis. The ACS and the flanker task are mature measures of attentional control. However, in the present study, the correlation coefficient is less than 0.3, indicating a significantly low relationship. These reported correlations would not survive Bonferroni correction for multiple comparisons. This suggests the low efficiency of spontaneous EEG as a measure of attentional control. This limits the reliability of the conclusions in the present study. However, the present study remains significant, as the results highlight the caveats and limitations associated with employing spontaneous EEG indicators as measures of attentional control in future studies. In future studies, attention should be directed towards the instability, influenced by factors such as eye opening and closing effects, as well as the overall low validity of spontaneous EEG indicators.

In summary, our study successfully resolved a long-standing research controversy by indicating that, in the eyes-open condition, the theta/beta power ratio did not reflect individuals’ attentional control ability but, rather, their beliefs about this attentional control ability. However, the EEG alpha oscillation may be related to attentional control ability.

## Figures and Tables

**Figure 1 behavsci-14-00227-f001:**
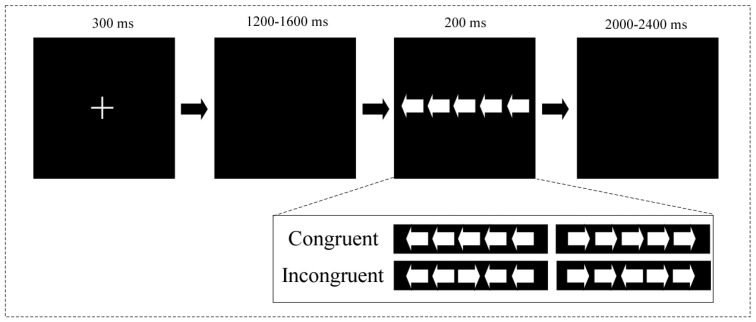
A schematic representation of the flanker task.

**Figure 2 behavsci-14-00227-f002:**
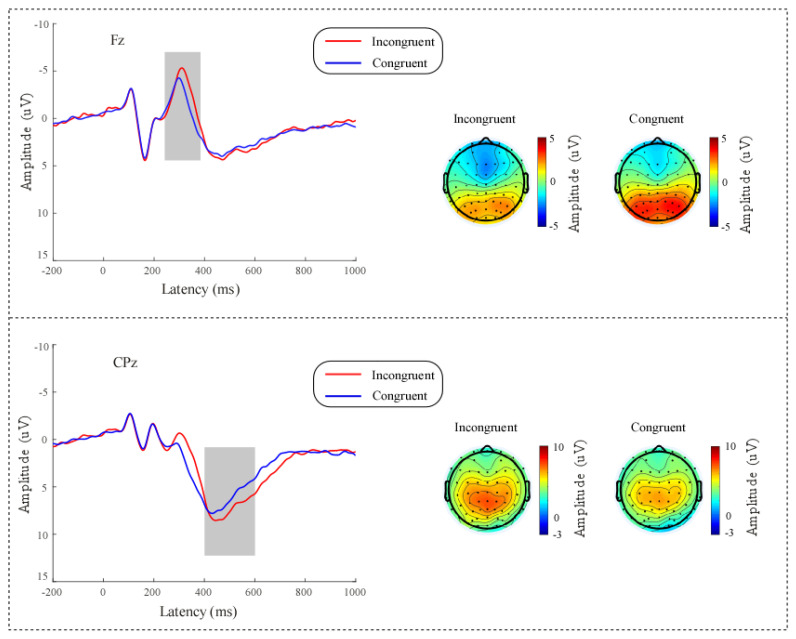
**Upper**: Grand means of the event-related potential (ERP) waveforms for congruent and incongruent conditions at electrode site Fz and topographies of the N2 amplitude (230–380 ms); **Lower**: Grand means of the ERP waveforms for congruent and incongruent conditions at electrode site CPz and topographies of the P3 amplitude (400–600 ms).

**Table 1 behavsci-14-00227-t001:** The mean (M) and standard error (SE) of spontaneous EEG indicators: theta/beta power ratio and alpha power.

Theta/Beta Power Ratio							Alpha Power					
Open				Close			Open			Close		
Fz		Cz	Pz	Fz	Cz	Pz	Fz	Cz	Pz	Fz	Cz	Pz
M	2.33	2.39	2.08	2.49	2.48	2.20	1.93	1.98	2.00	3.95	4.06	4.10
SE	0.069	0.062	0.056	0.080	0.082	0.082	0.079	0.080	0.095	0.230	0.236	0.277

Notes: Open = eyes-open condition; Close = eyes-closed condition.

**Table 2 behavsci-14-00227-t002:** The correlation between the theta/beta power ratio and subjective or objective attentional control measures.

		ACS	RT_Interference	N2d	P3d
Open_θ/β_Fz	*r*	−0.263	−0.092	−0.005	0.056
	*p*	0.046 *	0.491	0.97	0.677
Open_θ/β_Cz	*r*	−0.249	−0.015	−0.051	−0.035
	*p*	0.059	0.913	0.703	0.796
Open_θ/β_Pz	*r*	−0.263	0.058	−0.066	−0.027
	*p*	0.046 *	0.668	0.624	0.843
Close_θ/β_Fz	*r*	−0.223	−0.045	−0.055	0.053
	*p*	0.093	0.736	0.679	0.695
Close_θ/β_Cz	*r*	−0.238	−0.025	−0.082	0.010
	*p*	0.072	0.851	0.540	0.943
Close_θ/β_Pz	*r*	−0.246	0.047	−0.081	−0.037
	*p*	0.063	0.729	0.544	0.785

Notes: Open = eyes-open condition; Close = eyes-closed condition; θ/β = theta/beta power ratio; RTs = reaction times; ACS = attentional control scale; N2d = N2d amplitude; P3d = P3d amplitude; * *p* ≤ 0.05.

**Table 3 behavsci-14-00227-t003:** The correlation between the alpha power and subjective or objective attentional control measures.

		ACS	RT_Interference	N2d	P3d
Open_α_Fz	*r*	−0.131	−0.161	0.007	0.081
	*p*	0.326	0.228	0.956	0.548
Open_α_Cz	*r*	−0.086	−0.188	0.058	0.155
	*p*	0.522	0.156	0.667	0.245
Open_α_Pz	*r*	−0.096	−0.266	0.067	0.186
	*p*	0.474	0.044 *	0.618	0.162
Close_α_Fz	*r*	−0.046	−0.05	0.235	0.254
	*p*	0.734	0.711	0.076	0.054
Close_α_Cz	*r*	−0.055	−0.095	0.235	0.292
	*p*	0.682	0.478	0.076	0.026 *
Close_α_Pz	*r*	0.006	−0.166	0.220	0.285
	*p*	0.964	0.214	0.096	0.030 *

Notes: Open = eyes-open condition; Close = eyes-closed condition; α = alpha power; RTs = reaction times; ACS = attentional control scale; N2d = N2d amplitude; P3d = P3d amplitude; * *p* ≤ 0.05.

**Table 4 behavsci-14-00227-t004:** The correlation between the attentional control measures.

		ACS	RT_Interference	N2d	P3d
ACS	*r*	—			
	*p*	—			
RT_Interference	*r*	0.305	—		
	*p*	0.020 *	—		
N2d	*r*	0.204	0.092	—	
	*p*	0.124	0.493	—	
P3d	*r*	0.152	−0.261	0.413	—
	*p*	0.255	0.047 *	0.001 *	—

Notes: RTs = reaction times; ACS = attentional control scale; N2d = N2d amplitude; P3d = P3d amplitude; * *p* ≤ 0.05.

## Data Availability

Data are available upon reasonable request to the authors.

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
