# Peer review of "Can the Spontaneous Electroencephalography Theta/Beta Power Ratio and Alpha Oscillation Measure Individuals’ Attentional Control?"

_behavsci, 2024, doi:10.3390/bs14030227_

Round 1

Reviewer 1 Report

Comments and Suggestions for Authors

Review: The authors investigate the relationship between neural oscillations and attentional control score. While the article is easy to read, major concerns limit the interpretation and the contribution to the field.

Major concerns:

1.       The main findings from the study is that alpha oscillations are better predictors of attentional control. This has been shown in numerous studies in the past using multiple experimental setups. Thus, the manuscript neither adds to nor reject any theories about how neural oscillations play a role in attention control.  

2.       The authors use correlation has a main measure to test the relationship between the measures: theta/beta ratio and alpha power and ACS scores. While correlation analysis is useful in certain cases, the interpretation of such analysis has serious limitations and should be treated with caution. Here, while certain effects were statistically significant (p<0.05), the effect sizes for most of the correlations remain medium. This limits the interpretability of the core results.

3.       The discussion section could be strengthened to highlight the differences between the current study and previous studies that investigated the relationship between theta/beta power ratio and ACS score.  

Minor concerns:

1.       The authors should consider including a figure for the Flanker experiment.

2.       The authors should include a paragraph to show the relation between the English version of the ACS score and the Chinese adaption of the score to better inform the readers about the scale.  

Comments on the Quality of English Language

The English can be improved.

Reviewer 2 Report

Comments and Suggestions for Authors

Can the spontaneous electroencephalography theta/beta power ratio and alpha oscillation measure individuals’ attentional control?

This study aims to test the hypothesis that theta/beta ratio and alpha oscillation reflect different aspects of attentional control. The author measured RT in a flanker task as a measure of objective attentional control; scores on the attentional control scale (ACS) questionnaire as a measure of subjective attentional control belief; and the difference between congruent and incongruent trials in N2 and P3 ERP waves. Moreover, theta/beta power ratio and alpha oscillation were measured, both in eye open and eye closed conditions. The authors found evidence for the relation of theta/beta power ratio with attentional control belief and of alpha oscillation with attentional control ability.

Evaluation

The question regarding the spontaneous EEG markers of attentional control is interesting, and the idea to dissociate different oscillations with objective and subjective measures of attentional control is great. However, I have a couple of important concerns, both theoretical and empirical.

One concern is that the manipulation of eyes open and eyes closed is not well explained. There is no theoretical justification for it apart from saying that there is evidence for better results in previous studies, with eye closed. Likewise, the discussion doesn’t include any theoretical justification for the different results with eyes open and closed. Moreover, when trying to explain the discrepancy with other studies, the authors suggest that it may be a result of not discriminating between eyes open and closed but without providing evidence for this claim.

Second, while the dissociation between objective and subjective measures of attentional control is interesting and important, and the idea to test both alpha and theta/beta ratio is relevant, the results don’t really match with the expectations. If N2d and P3d should be a measure of objective attentional control ability, how is the lack of interaction with RTs explained? Importantly, why would alpha oscillations be correlated positively with the ERP marker but negatively with the RT marker of the interference? This is a great example of a paper that would benefit from pre-registration, as there are clear expectations. Which brings me to the last point.

Third, what are the expectation? What are the predictions? Please state them at the end of the introduction.

Finlay, CPz is used as a reference for recording, but what is the reference for analysis? You need an electrode that doesn’t produce any data. I must have missed it but if I did, please clarify.  

 I have additional comments which I summarized below, mainly at the order of their appearance.

 Introduction:

 1.      Page 2, line 76-77, the sentence “Although studies have revealed a correlation between alpha oscillations in spontane-76 ous EEG and attention activity, no clear evidence currently supports a correlation between 77 spontaneous EEG alpha activity and attentional control processes [21-23].” Is not very clear. Do you mean between attentional ability and attentional control ability? Please clarify.

2.      I already mentioned before, but on page 2 it is evident. Apart from the magnitude, what is the theoretical reason to use eyes open and eyes closed?

3.      The last paragraph of the introduction should be “in summary” rather than “above all”, otherwise it is a repetition.  

4.      What are the predictions? Please finish the introduction with clear predictions to all the experimental conditions.

 Materials and Methods:

 5.      Please explain how you got to the number of participants. Is there any power analysis to determine that?

6.      Section 2.2.2, large parts of this section belong to the procedure rather that the description of the measure. For example lines 139-144, and 147-151.

7.      Experimental procedure, it says that there are four steps but only three are described.

8.      AFz and CPz are used as ground and reference for data collection, but what is the reference for the analysis?

9.      Please give justifications for your choices. For example, why only three electrodes (Fz, Cz, Pz)? Why do you apply square root transformation to the spontaneous EEG?

 Results:

 10.  You only report correlations. Please provide the alpha and theta/beta power ratio.

11.  The results are significant but mostly just reached significance. I am a bit concerned about it. Please discuss. Do you have enough power?

 Discussion

 12.  On page 8, line 306 you say “The inconsistent and unexplained results in previous research may be largely attributed to the failure to differentiate between eyes-open and eyes-closed conditions.” However you don’t provide any evidence for this. It is a relatively easy task to do. Have the experiments that didn’t find the correlation done with eyes open? Closed? This is an interesting idea but it has to be proved.

Comments on the Quality of English Language

English is fine. Minor editing is required, mainly regarding punctuation. 

Reviewer 3 Report

Comments and Suggestions for Authors

This manuscript investigates TBR and alpha waves' contribution to attentional control. The authors claim this to be the first study that separates eyes open and eyes closed conditions while conducting EEG analysis. 

However, there are several studies which have used similar approach for eg. see (https://www.sciencedirect.com/science/article/pii/S0278584610003076). Authors have not even cited this paper. How do authors explain their results compared to results shown in this paper?

The other concern is regarding statistics, authors have performed multiple comparisons with doing any post hoc test. The reported p -values are ~0.04s. My guess is that significance of the test results may not hold true if the stats is applied appropriately. The big claim in the last paragraph in discussion is based on very weak statistics.

Comments on the Quality of English Language

Please get the manuscript proof reading by a native English speaker

Reviewer 4 Report

Comments and Suggestions for Authors

The paper is interesting, and well-written.

However, there are some concerns to be addressed before the publication:

1.      It is not clear from the material and methods if the authors are using all the electrodes for the analysis, or they are diving them based on the location.

2.      Are the authors using some correction for the multiple comparisons? Such as FDR or Bonferroni?

3.      The versions for EEGlab and JASP used for the analysis should be reported in the manuscript.

4.      In Figure 1, it would be nicer adjusting the y axis of the spectrogram based on the minimum value. For example, in Fz the y axis should start from 4. In this way, the figure will be clearer, and the readers may appreciate more the different bands.

Round 2

Reviewer 1 Report

Comments and Suggestions for Authors

I appreciate the authors’ efforts to address some of the concerns raised in my previous review. However, the response does not adequately address the fundamental issue that the overall effects are still not compelling.

Could the authors try implementing additional strategies/analysis to enhance the impact of the article?

Comments on the Quality of English Language

The article is written well. 

Author Response

Thanks for your comments. And we understand that you may have concerns regarding the modest correlations. As advised by the academic editor, we have explicitly acknowledged this limitation in the revised manuscript (lines 406-407). These modest correlations would not survive bonferroni correction for multiple comparisons. Of course, if you have further specific data processing suggestions, we are more than willing to use them to make modifications.

Reviewer 2 Report

Comments and Suggestions for Authors

I am satisfied with the authors response to me comments. 

Author Response

Thanks for your positive comments.

Reviewer 3 Report

Comments and Suggestions for Authors

I had my concerns with the statistics used in the manuscript, which the authors agree to and have also mentioned this in the discussion now. I would like to leave the final decision on the editor regarding this limitation.

Author Response

Thanks for your comments. And we understand that you may have concerns regarding the modest correlations. As advised by the academic editor, we have explicitly acknowledged this limitation in the revised manuscript (lines 406-407). These modest correlations would not survive bonferroni correction for multiple comparisons.

Round 3

Reviewer 1 Report

Comments and Suggestions for Authors

Perhaps the authors can consider increasing the number of subjects in the study, if that is possible.

Author Response

Thank you, we appreciate the value of your suggestion. However, we were constrained by two primary reasons. Firstly, G*Power recommended a minimum total sample size of 45 for correlation analysis. We invited 58 participants to partake in our study, thus exceeding this requirement. Secondly, the time limitations we faced posed a challenge in increasing the number of subjects. Consequently, we opted not to implement this suggestion.

Round 4

Reviewer 1 Report

Comments and Suggestions for Authors

1. Can the authors explain and elaborate on why they chose a target effect size of 0.395, this effect size is rather small ? 

Author Response

Thanks for your comments. In previous research (Putman, P., et al., EEG theta/beta ratio in relation to fear-modulated response-inhibition, attentional control, and affective traits. Biological Psychology, 2010. 83(2): p. 73-78.), individuals’ theta/beta power ratios were inversely correlated with their ACS scores (r = -0.395). This study proved that the theta/beta power ratio may be related to attentional control. Thus, we chose a target effect size of 0.395. And, related to your previous suggestion. If a larger effect size is selected, the calculation will necessitate a smaller sample size. This observation further underscores the adequacy of the sample size in our study.